# Effects of Distributional Biases on Gradient-Based Causal Discovery in the Bivariate Categorical Case

## Abstract

Gradient-based causal discovery shows great potential for deducing causal structure from data in an efficient and scalable way. Those approaches however can be susceptible to distributional biases in the data they are trained on. We identify two such biases: *Marginal Distribution Asymmetry*, where differences in entropy skew causal learning toward certain factorizations, and *Marginal Distribution Shift Asymmetry*, where repeated interventions cause faster shifts in some variables than in others. For the bivariate categorical setup with Dirichlet priors, we illustrate how these biases can occur even in controlled synthetic data. To examine their impact on gradient-based methods, we employ two simple models that derive causal factorizations by learning marginal or conditional data distributions – a common strategy in gradient-based causal discovery. We demonstrate how these models can be susceptible to both biases. We additionally show how the biases can be controlled. An empirical evaluation of two related, existing approaches indicates that eliminating competition between possible causal factorizations can make models robust to the presented biases.

## 1 Introduction

Causal discovery is an increasingly popular research field since causal understanding enables more informed decision-making and potentially better generalization to new settings and data distributions. Conditional distributions of data can be expressed by Bayesian networks in the form of

$$P(X_1, \ldots, X_n) = \prod_{i=1}^{n} P(X_i | Pa_{X_i}). \tag{1}$$

Eq. 1 expresses that each variable $X_i$ depends on a set of parents $Pa_{X_i}$ (this might also be the empty set) and, conditioned on these parents, is independent of other variables. This is called a factorization and can be represented as a directed acyclic graph (DAG), which has variables as vertices, while edges express dependence relations. When the edges express causal dependence (meaning that the state of $X_i$ is caused by the states of parents $Pa_{X_i}$), this is called a structural causal model (SCM) (Pearl, 2009; Schölkopf et al., 2021).

The simplest yet meaningful causal setup to investigate is that of only two, fully observed, variables. In this case, if a causal relationship is assumed, data can either be generated through a process $P(X_1, X_2) = P(X_1)P(X_2|X_1)$, or $P(X_1, X_2) = P(X_2)P(X_1|X_2)$. In the first case, $X_1$ is the independent variable upon which $X_2$ depends, and in the second case, it is the other way around. As an example, consider the relationship between weather and outfit choice: Sunny and warm weather will see people dress differently than rainy or cold weather. From observational data alone, the causal relationship of the two variables cannot generally be learned. Interventions are required to determine a causal effect (Pearl, 2009). In our example, a change in weather will see people change their outfits, while a change in outfits will unfortunately not bring the sun out.

But how to derive these causal relations from data? Recent works improve upon discrete score-based methods by using gradient-based approaches with differentiable score functions and models. These models are commonly based on three components: (i) Neural networks with a capacity to learn distributions that relate variables to their parents, (ii) a differentiable representation of the graph structure between the variables,

and (iii) a regularization or constraint that enforces that the factorization represented by these distributions is a valid DAG. These methods are trained on observational or interventional data, with known or unknown interventions. They reconstruct input data based on conditional distributions, given what the model believes are causal parents, and optimize a certain score function – commonly a maximum likelihood loss with additional regularizations enforcing sparseness and DAGness (Bengio et al., 2020; Brouillard et al., 2020; Lippe et al., 2022; Ke et al., 2023).

In this context, the regularization means that models have to decide between possible parents for variables under the constraint of achieving a DAG structure. The distributions and factorizations to which the models converge, based on their loss, are clearly influenced by differences and changes (due to interventions) in the data distributions. As we show in this paper, such influences lead to two types of biases in joint and marginal distributions: *Bias 1: Marginal Distribution Asymmetry* and *Bias 2: Marginal Distribution Shift Asymmetry*.

We define Bias 1 as the difference in entropy between marginal distributions and show how it can be controlled, in the bivariate categorical setup, by parameterized deviations from a Bayesian Dirichlet equivalence (BDe) prior. Likewise, we define Bias 2 as the difference of Kullback-Leibler Divergences between distributions before and after interventions. We show how it can be controlled by adjusting the relative frequency of interventions performed on the individual variables. These biases have the potential to significantly aid or hinder correct learning of causal structure depending on the choice of model and training paradigm. To the best of our knowledge, these biases – in particular for the setup of continuing interventions – have not been investigated in the literature before.

In this work, we empirically show and discuss the influence of both biases on recovering causal structure using two different simple models. We choose the common bivariate causal discovery problem with categorical data; a simple setting that serves as a clear and illustrative example. For the same reason, it is also a common distributional choice in related works, e.g. Bengio et al. (2020). Specifically, our **contributions** are:

1. We present two distributional biases that can affect convergence in gradient-based causal discovery methods: *Bias 1: Marginal Distribution Asymmetry* and *Bias 2: Marginal Distribution Shift Asymmetry*. We prove the existence of *Bias 2* for the scenario where interventions are performed exclusively on the causal (parent) variable.

2. We demonstrate how Bias 1 can be controlled through deviations from a BDe prior in the bivariate categorical case, and how Bias 2 can be controlled by adjusting the relative frequencies of interventions.

3. We propose two simple models to empirically examine how these distributional biases affect gradient-based learning of marginal or conditional distributions, and consequently, causal factorizations. Using these models, we demonstrate the practical impact of both biases in a controlled bivariate categorical setting.

4. We additionally evaluate two existing causal discovery approaches for the bivariate categorical setup (Bengio et al., 2020; Lippe et al., 2022), and find that the models with direct competition between possible factorizations are susceptible to the biases, while the model without this competition is not.

Overall, our study demonstrates how distributional biases in data – particularly in artificially generated data – can influence gradient-based causal discovery, highlighting the need to consider such biases in causal inference methods.

## 2 Related work

Discovering the graph structure of Bayesian Networks from samples of their joint distribution is called structure learning. If the network represents cause-effect relationships between the variables, it is called causal discovery. Approaches to this problem are either constraint-based or score-based.

Constraint-based methods try to recover the true causal graph by exploiting conditional independence between the variables (Monti et al., 2019; Spirtes et al., 2000; Kocaoglu et al., 2019; Jaber et al., 2020; Sun et al.,

2007; Hyttinen et al., 2014; Peters et al., 2016; Heinze-Deml et al., 2018). That is, they perform exhaustive statistical tests of samples from subsets of variables. Identified independencies are used to remove causal edges between variables to converge to an estimate of the causal structure. The downside of constraint-based methods is their computational complexity and poor scaling with respect to the number of variables.

Score-based methods, on the other hand, optimize for a certain score (e.g. Bayesian Information Criterion) to recover the causal structure. Those scores represent how well the discovered structure can model the data, and incorporate further constraints (i.e. DAGness) and penalties (e.g. for the total number of edges in the found graph). Traditionally, score-based methods search discretely through the space of possible graph structures. Given the super-exponential number of such graphs, heuristic search methods need to be applied (Tsamardinos et al., 2006; Meek, 2023; Hauser & Bühlmann, 2012; Wang et al., 2017; Yang et al., 2018).

To eliminate the need for combinatorial search through graph structures in score-based methods, more recently, continuous methods for causal discovery and structure learning have been proposed. Those approaches use differentiable score functions as well as differentiable causal models, where the functions predicting the causal variables given their parents are often modeled using neural networks and a continuous relaxation of an adjacency matrix representing the causal relationships between variables. Due to the differentiability of the system, modern gradient-based methods can be applied. In this context, Zheng et al. (2018) introduce a smooth constraint in learning a linear SCM that enforces the learned graph to be a DAG, an approach which has been extended to the nonlinear case (Yu et al., 2019; Lachapelle et al., 2020). Wren et al. (2022) aim to learn the true graph using discrete backpropagation and the same regularization as Zheng et al. (2018) to enforce DAG-ness. Zhu et al. (2020) uses Reinforcement Learning to find the correct causal graph structure. However, these methods only consider observational data, and hence can only recover graph structures in the same Markov equivalence class as the true causal graph.

Several other approaches do take interventions into account. Bengio et al. (2020) propose to learn the causal direction of two random variables, where one causes the other, by a meta-learning approach. First, they independently learn models for all possible causal factorizations on a single observational distribution, which is followed by joint learning of those model parameters alongside structural variables that encode the causal direction on several interventional distributions. The authors show both empirically and analytically that the structural parameters converge to the correct causal direction since this model requires updating fewer parameters to adapt to a new interventional distribution.

The work of Bengio et al. (2020) is extended to more than two causal variables by SDI (Ke et al., 2023). In this approach, the conditional and prior distributions are similarly modeled by MLP parameters $\theta$, while the causal structure is modeled by structural parameters $\gamma$. The model is trained in an alternating scheme, where the functional parameters are fitted using the observational distribution, while the structural parameters are fitted using interventional distributions. Each variable is represented by a single MLP, enabling the approach to scale linearly with respect to the number of causal variables. As an enhancement to SDI, Lippe et al. (2022) propose ENCO, which uses a new gradient estimator for the structural parameters and splits them into parameters for edge existence and edge direction. A related approach for continuous distributions has also been proposed (Brouillard et al., 2020). Ke et al. (2020) propose a model that is trained on many different causal graphs and can predict new causal graphs given interventional samples, where the intervention target is known. Their approach uses an attention mechanism to decide which variable is used to which extent to predict another. Ng et al. (2022) learn a binary adjacency matrix representing the causal structure. They use a Gumbel-Sigmoid to threshold the values of the adjacency matrix and approximate a discrete distribution. However, they do not consider interventions but learn solely from observational data in cases where the true causal graph is identifiable up to a super-graph. For a comprehensive summarization of continuous optimization methods for causal discovery, we refer to the review by Vowels et al. (2022).

Reisach et al. (2021) highlight how varsortability – the tendency of marginal variances to increase along causal directions in certain linear additive noise models – can inadvertently leak information about causal structure. They demonstrate that typical random parameter choices in these models yield "varsortable" data that artificially boosts performance for structure learning methods, even in purely observational scenarios. Likewise, in our bivariate categorical setup, we also observe that imbalanced Dirichlet priors introduce

asymmetric observational distributions, which steer gradient-based algorithms toward one causal direction without needing explicit causal constraints. Unlike their work, we investigate the interventional setting.

## 3 Biases

We identify two biases for the fully observable bivariate causal setup. In this section, we define both biases before we empirically validate them in subsequent sections on a testbed of two categorical variables with Dirichlet priors.

### 3.1 Marginal Distribution Asymmetry

While it is, in principle, impossible to determine causality from observational data alone (Pearl, 2009), a causal relationship between variables often results in asymmetries between marginal distributions $P(X_1)$ and $P(X_2)$ even in the observational case, before an intervention has taken place (Mooij et al., 2016). This leads to a bias we call *Bias 1: Marginal Distribution Asymmetry*. We define it as follows:

**Definition 1** (Bias 1: Marginal Distribution Asymmetry). *We define the difference in entropy between random variables $X_i$, $X_j$ as*

$$\Delta H_{i,j} = H(X_i) - H(X_j).$$

*We say there is a* marginal distribution asymmetry *when $\Delta H_{i,j} \neq 0$.*

An asymmetry in distributions can lead to an asymmetry in the learning signals to a gradient-based causal discovery method. In the bivariate categorical case we find that the spikier distribution with $\Delta H_{i,j} < 0$ is easier to learn. Distribution asymmetry can be controlled for by a suitable choice of conditional $P(X_j|X_i)$, at least in a synthetic setup. In the categorical case with Dirichlet prior, we present the Bayesian Dirichlet equivalence prior (see Section 4.1) as an example. For clarity, we provide the definition for (conditional) entropy in Section A.1 of the Appendix.

### 3.2 Marginal Distribution Shift Asymmetry

An intervention in a causal setup is commonly applied by fixing the distribution of a random variable (to a specific value for hard interventions or a distribution over values for soft interventions Pearl (2009)) and observing how this affects marginal distributions of other variables. Such interventions lead to shifts in distributions, which generally happen at different speeds for different causally related variables. Unrelated variables do not change at all (see ICM principle in Schölkopf et al. (2021)). We call the resulting bias *Bias 2: Marginal Distribution Shift Asymmetry* and define it as follows:

**Definition 2** (Bias 2: Marginal Distribution Shift Asymmetry). *Let $X_i \sim P_i, X_j \sim P_j$ be two random variables, and $X_i' \sim P_i', X_j' \sim P_j'$ the same random variables after an intervention. Then, we define the distribution shift[1] of variable $X_i$ with the Kullback-Leibler divergence as*

$$S_i = D_{KL}(P_i'||P_i).$$

*The difference in distribution shift between variables $X_i$ and $X_j$ is then*

$$\Delta S_{i,j} = S_i - S_j.$$

*We say there is a* marginal distribution shift asymmetry *when $\Delta S_{i,j} \neq 0$.*

The definition of (conditional) Kullback-Leibler divergence is likewise provided in Section A.1 of the Appendix.

In a bivariate setup with factorization $X_1 \rightarrow X_2$ (i.e. $X_1$ causes $X_2$) there are four intervention cases to differentiate. The reason is that interventions momentarily change dependencies when fixing variables (e.g. making $X_2$ independent of $X_1$), and $\Delta S_{i,j}$ depends on causal dependencies before and after an intervention. We visualize these cases in Figure 1 and outline their associated distribution shifts in Table 1 for marginal

---

[1]An alternative formulation based on Cross Entropy is given in Appendix A.3

as well as conditional distributions. Appendix Section A.4 contains an empirical verification on bivariate categorical data. We assume continuous interventions, where a new intervention is applied immediately upon undoing a previous intervention. Some other works undo interventions by restoring the original distributions first. This, however, is a special case of the more general continuous interventions.

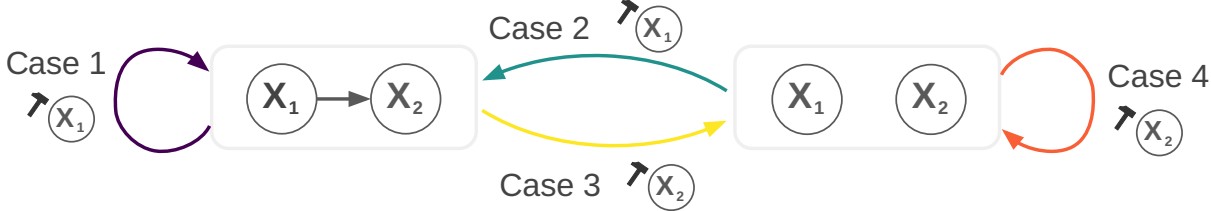

Figure 1: Bivariate intervention cases for causal factorization $X_1 \to X_2$, color-coded. An intervention, signified by the little hammer, assigns a new independent distribution to a variable. The system can therefore have two states: The default causally related one (left box), and a causally independent one (right box), which is obtained by intervening on $X_2$. In case 2, the underlying conditional $P(X_2|X_1)$ is restored.

Table 1: Different intervention cases and their effect on marginals and conditionals. The **M** row describes behavior of marginal distributions, the **C** row that of conditionals. Causal relationships in the assumed causal setup $X_1 \to X_2$ might not be present, if $X_2$ was fixed through an intervention. Corresponding empirical data of distribution shifts, for the bivariate categorical setup, is provided in Section A.4 of the Appendix.

|   | Case 1 ● | Case 2 ● | Case 3 ● | Case 4 ● |
|---|---|---|---|---|
| **M** | $P_1$ changes arbitrarily. $P_2$ can change at most as much as $P_1$ (see Theorem 1 in Appendix). | $P_1$ changes arbitrarily. $P_2$ also changes arbitrarily as it was arbitrary before intervention. | $P_1$ remains unchanged as it is independent. $P_2$ changes arbitrarily. | $P_1$ remains unchanged as it is independent. $P_2$ changes arbitrarily. |
| **C** | $P_{2|1}$ stays constant, $P_{1|2}$ changes arbitrarily. | $P_{2|1}$ changes arbitrarily, as does $P_{1|2}$. | $P_{2|1}$ always changes more than $P_{1|2}$. | $P_{2|1}$ changes arbitrarily. $P_{1|2}$ stays constant. |

It follows that the order of interventions matters. In practice, it is often desirable to intervene in a random order to avoid exacerbating the distribution shift bias.

A bias in the distribution shifts under intervention generally results in an asymmetric learning signal, because the loss landscape changes more rapidly for the variable with the larger distribution shift. The distribution shift bias can be controlled by empirically determining an appropriate ratio of interventions on $X_1$ compared to $X_2$. To a limited extent, it can also be controlled by choosing a specific order of interventions to make either case 1 or case 2 more frequent.

## 4 Methods

We investigate the problem of continuous score-based causal discovery from a similar perspective as recent related work (Bengio et al., 2020; Ke et al., 2023). That is, we assume access to samples of discrete random variables $X_1$, $X_2$ that are causally related without any further confounders, and try to recover the causal direction between them. We repeatedly intervene on the distributions of these variables ($I$ times), giving rise to a set of different, related distributions $S = \{P_i(X_1, X_2)\}_{i=1}^{I}$, called the interventional set.

We assume a differentiable score function, which quantifies how well a given (differentiable) model can predict samples from the distributions in $S$. The model can then be trained using gradient-based learning to optimize the given score function. The model is constructed in such a way that optimizing the score leads to a given factorization of $P(X_1, X_2)$ that aims to reflect the true causal structure.

### 4.1 Bivariate Categorical Setup

For our investigation, we consider categorical distributions, which are themselves sampled from a Dirichlet prior. The Dirichlet prior defines a distribution over probabilities for different categories. It is the conjugate prior to categorical distributions and a natural choice, not only because it always returns normalized probabilities, but also because it is versatile given its parameters.

**Categorical distributions with Dirichlet prior** We consider categorical random variables $X_1$, $X_2$, with $K$ categories each, respectively, defined by probability vectors $\boldsymbol{\pi}_1 \sim \text{Dirichlet}(K, \boldsymbol{\alpha}_1)$, $\boldsymbol{\pi}_2^{X_1=x_1} \sim \text{Dirichlet}(K, \boldsymbol{\alpha}_2)$, which are random variables themselves. Note that the different probability distributions $\boldsymbol{\pi}_2^{X_1=x_1}$ are sampled independently for each value of $X_1$ but from the same Dirichlet distribution. The distributions of $X_1$ and $X_2$ are then given as

$$P(X_1) = \text{Categorical}(\boldsymbol{\pi}_1) \tag{2}$$

$$P(X_2|X_1 = x_1) = \text{Categorical}(\boldsymbol{\pi}_2^{X_1=x_1}) \tag{3}$$

That is, we use Dirichlet priors to generate categorical distributions for both variables. The Dirichlet distributions, in turn, are specified by parameter vectors $\boldsymbol{\alpha}_1$ and $\boldsymbol{\alpha}_2$ that determine their shape. We set these parameters to

$$\boldsymbol{\alpha}_1 = \mathbf{1}_K \tag{4}$$

$$\boldsymbol{\alpha}_2 = \frac{1}{\varepsilon K} \, \mathbf{1}_K \tag{5}$$

where $\mathbf{1}_K$ is a $K$-dimensional vector of ones and $\varepsilon$ a parameter to control Bias 1.

Samples $x_1, x_2$ are generated using either ancestral sampling or independent sampling, depending on the interventional context. Under ancestral sampling, a value $x_1$ is first drawn from the marginal distribution $P(X_1)$, after which a subsequent value $x_2$ is sampled from the conditional distribution $P(X_2 \mid X_1 = x_1)$. Alternatively, after intervention cases 2 or 4, $x_1$ and $x_2$ become momentarily independent and both variables are sampled independently from respective marginal distributions.

**The BDe prior** To achieve symmetry of marginal distributions in our generated synthetic data, at least for an observational case, we borrow the Bayesian Dirichlet equivalence (BDe) prior from the classical literature of structure learning (Koller & Friedman, 2009, Section 18.3.6.3). This allows us to control the extent of *Bias 1: Marginal Distribution Asymmetry* and investigate its effects in isolation. It requires that the values of all $\boldsymbol{\alpha}$ vectors of the various Dirichlets given different possible states $x$ of $Pa_{X_i}$ add up to the same total value $\sum_x \boldsymbol{\alpha}_i^{Pa_{X_i}=x}$, for all variables $X_i$. This sum is called the equivalent sample size, and the prior ensures that all Dirichlet distributions have the same equivalent sample size. It is used in structure learning as an initial unbiased assumption on the distribution of Dirichlet-distributed parameters; in fact, the only one that ensures equivalent scores for possible factorizations (Koller & Friedman, 2009, Theorem 18.4).

To obtain unbiased distributions, we make use of this same principle for data generation. In our case, a BDe prior is ensured by the factor $1/K$ in Eq. (5), because $X_1$, the parent of $X_2$, can take $K$ possible values. The factor $\varepsilon$ is used to deviate from the BDe equivalence scenario in a controlled fashion and investigate its effect. The BDe prior is obtained if $\varepsilon = 1$. If $\varepsilon > 1$, $X_1$ has, on average, a higher entropy, or $P(X_1)$ is closer to a uniform distribution, than $X_2$. If $\varepsilon < 1$, the opposite is true.

Using the marginalized probability $P(X_2) = \sum_{x_1} P(X_2|X_1 = x_1)P(X_1 = x_1)$ and the difference in entropy $\Delta H_{1,2} = H(X_1) - H(X_2)$ from Definition 1, this means that for our particular setup:

$$\mathbb{E}[\Delta H_{1,2}] = \begin{cases} > 0, & \text{if } \varepsilon > 1, \\ 0, & \text{if } \varepsilon = 1, \\ < 0, & \text{if } \varepsilon < 1. \end{cases}$$

In other words, $\varepsilon = 1$ yields symmetric marginals (in terms of entropy) on average (no bias), while $\varepsilon > 1$ or $\varepsilon < 1$ biases the entropies of $X_1$ vs. $X_2$. Figure 3b in Section 5 confirms this empirically.

**Interventions** Similar to Bengio et al. (2020), we consider soft interventions on the causing variable $X_1$. We also consider soft interventions on the dependent variable $X_2$, and a mix of interventions on both variables at varying rates.

An intervention on $X_1$ means sampling a new $\boldsymbol{\pi}_1$ for $P(X_1)$, while leaving the conditional $P(X_2|X_1)$ unchanged. The marginal $P(X_2)$ thus also changes. An intervention on $X_2$ means that $X_2$ is fixed to a new distribution and made independent of $X_1$. In practice, we sample a $\boldsymbol{\pi}_2$ from Dirichlet prior $\text{Dirichlet}(K, \boldsymbol{\alpha}_1)$ using the same prior as for $P(X_1)$ here. $P(X_1)$ remains unchanged. A later intervention on $P(X_1)$ will release the fix of $X_2$ and restore the conditional $P(X_2|X_1)$, which remains unchanged.

To control the *Bias 2: Marginal Distribution Shift Asymmetry*, we vary the ratio at which we intervene on $X_1$ and $X_2$. For this, we introduce a parameter $\lambda \in [0, 1]$ as the proportion of interventions carried out on variable $X_2$. That means when $\lambda = 0$, we only intervene on $X_1$, and when $\lambda = 1$ we only intervene on $X_2$. The fraction of interventions on $X_2$ increases linearly as $\lambda$ moves from 0 to 1. Interventions are always carried out in random order. Figure 3a in Section 5 shows the empirical relationship between $\lambda$ and $\Delta S_{1,2}$ for this setup. Figures 12, 13 and 14 in Appendix section A.4 present a further analysis of how different intervention cases contribute to $\Delta S_{1,2}$ for different $\lambda$.

## 4.2 Models

The goal of determining the correct causal direction between $X_1$ and $X_2$ is achieved by the surrogate goal of learning the distributions (conditional or marginal) of $X_1$, $X_2$ from samples across interventional distributions.

We employ two illustrative models: A *marginal model (MM)* that can learn either marginal distribution $P(X_1)$ or $P(X_2)$, and a conditional model (CM) that can learn either conditional distribution $P(X_1|X_2)$ or $P(X_2|X_1)$. Both models are depicted in Figure 2. They have structural parameters $c_1$ and $c_2$, which are weights that have to sum to one. These structural parameters encode how much the model prefers learning one distribution over the other.

Both models serve the purpose to investigate how *Bias 1: Marginal Distribution Asymmetry* and *Bias 2: Marginal Distribution Shift Asymmetry* affect convergence towards one (marginal or conditional) distribution or the other learnable, i.e., one causal factorization or the other. In other words: Which factorization is easier to learn under each bias?

**Marginal Model (MM)**

The marginal model only has one vector $\mathbf{i} \in \mathbb{R}^K$ of learnable parameters:

$$\hat{\mathbf{x}}_1 = c_2 \mathbf{i} \tag{6}$$
$$\hat{\mathbf{x}}_2 = c_1 \mathbf{i} \tag{7}$$

This model does *not* consider any input. However, it still learns through its loss and is at most able to learn the marginal distribution of either $X_1$ or $X_2$. The order of factors $c_i$ is chosen for consistency with CM, where a large $c_i$ corresponds to $X_i$ being the independent variable.

**Conditional Model (CM)**

The model has a learnable matrix $\mathbf{W} \in \mathbb{R}^{K \times K}$ that is multiplied with inputs:

$$\hat{\mathbf{x}}_1 = c_2 \mathbf{W} \, \mathbf{e}_{x_2} \tag{8}$$
$$\hat{\mathbf{x}}_2 = c_1 \mathbf{W} \, \mathbf{e}_{x_1} \tag{9}$$

The CM model is thus able to condition outputs on inputs, represented as one-hot encoded representations $\mathbf{e}_{x_i}$ of data values $x_i$. It is therefore able to learn a conditional distribution in $\mathbf{W}$.

**MM**

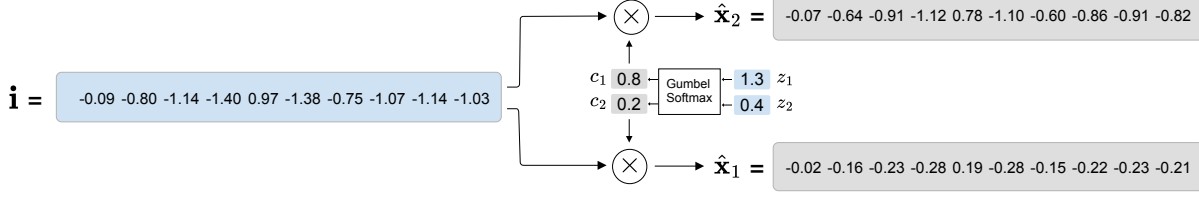

**CM**

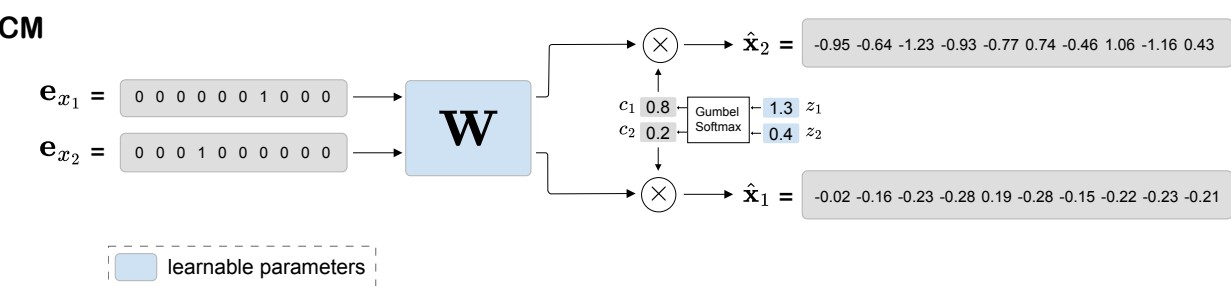

Figure 2: Exemplary visualization of the MM model (top) and CM model (bottom) from Section 4.2.

In both models, the structural parameters $c_1$ and $c_2$ are calculated from learnable logits $z_1, z_2$, given a temperature hyper-parameter $\tau$, as

$$c_1, c_2 = \text{Gumbel-Softmax}(z_1, z_2; \tau) \tag{10}$$

The Gumbel-Softmax is a variation of the softmax function where noise sampled from a Gumbel distribution is added to the logits $z_1, z_2$ before evaluating their softmax values $c_1, c_2$. The Gumbel-Softmax is commonly used as an approximation to represent samples from a discrete distribution in a differentiable way (Jang et al., 2017). The noise introduced by the Gumbel-Softmax serves the role of exploring both possible directions while raw logits are not too different, in order to not get stuck in one (possibly wrong) direction initially (Brouillard et al., 2020). From a causal discovery point of view, the structural parameters $c_1, c_2$ can be viewed as the causal factorization learned by each model: $c_1 > 0.5$ or $c_1 < 0.5$ denotes the case where $X_1$ causes $X_2$ or vice versa.

Details about initialization and hyperparameter values are provided in Appendix A.5.

**Training Paradigm**

The models are trained to minimize cross-entropy between predicted and ground-truth values of batches (of size $B$) from an observational or interventional distribution:

$$\mathcal{L} = -\frac{1}{B} \sum_{j=1}^{B} \sum_{i=1}^{2} \mathbf{e}_{x_i,j} \cdot \log(\text{softmax}(\hat{\mathbf{x}}_{i,j})) \tag{11}$$

For the factorization presented in Eq. (1), the joint log-likelihood is equivalent to the sum over conditional log-likelihoods of individual variables. If the $\hat{\mathbf{x}}_i$ can learn conditional distributions, $\mathcal{L}$ will thus minimize joint log-likelihood. In our MM model, and generally in the absence of input to condition on, however, $\hat{\mathbf{x}}_i$ can at most represent marginal distributions, meaning that $\mathcal{L}$ treats variables as independent without parents.

# 5 Results

We explore how Bias 1 and Bias 2 influence the behavior of the MM and CM models to converge to the correct factorization $X_1 \rightarrow X_2$ ($c_1 = 1$) or to the incorrect factorization $X_2 \rightarrow X_1$ ($c_1 = 0$). We investigate the observational as well as the interventional case and whether related work is also susceptible to the Biases.

For all following experiments on the convergence behavior of the models, 100 independent runs were conducted to gather expressive statistics. $K$ is set to 5. We always perform interventions in random order. For the related approach of Bengio et al. (2020) and Lippe et al. (2022), the corresponding implementation was used[2][3]. For reproducibility of our experiments, our code is available online[4].

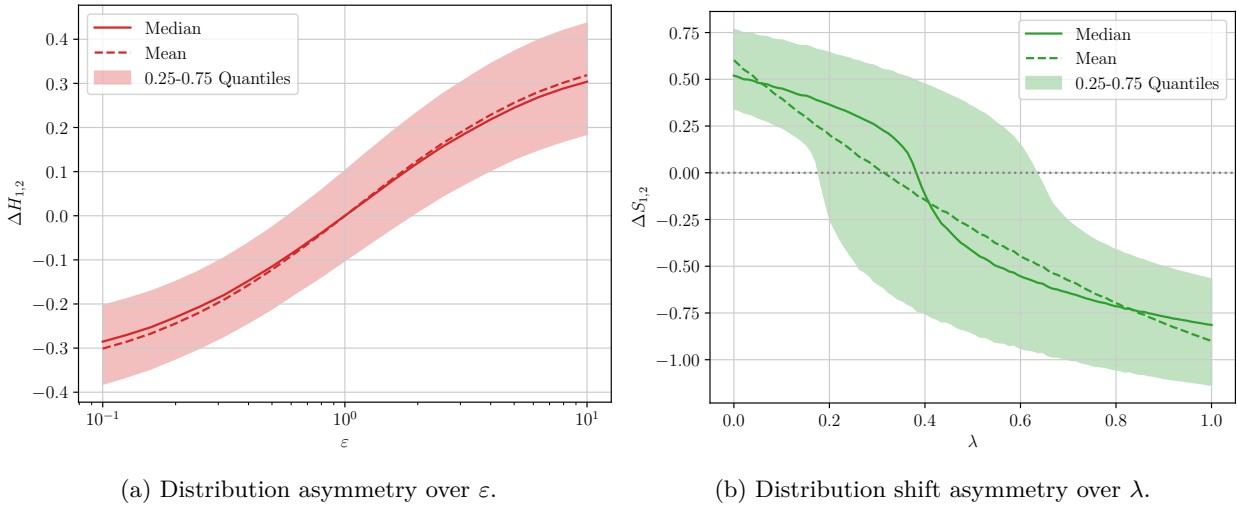

(a) Distribution asymmetry over $\varepsilon$.      (b) Distribution shift asymmetry over $\lambda$.

Figure 3: Bias 1 (Section 5.1) and Bias 2 (Section 5.2) for different values of $\varepsilon$ and $\lambda$ on the bivariate categorical setup with Dirichlet priors. Interventions happen in random order in Figure 3b.

## 5.1 Bias 1: Marginal Distribution Asymmetry

**Behavior with changing $\varepsilon$** Figure 3a shows the behavior of Bias 1 under changes of $\varepsilon$. It shows that for $\varepsilon = 1$, $\mathbb{E}\left[\Delta H_{1,2}\right]$ is $= 0$, i.e., the bias does not exist, and the distributions of $X_1$ and $X_2$ are symmetric. For $\varepsilon < 1$, $\mathbb{E}\left[\Delta H_{1,2}\right]$ is negative and the entropy of $X_1$ is smaller than that of $X_2$. For $\varepsilon > 1$, $\mathbb{E}\left[\Delta H_{1,2}\right]$ is positive and the entropy of $X_2$ is smaller. This demonstrates that the $\varepsilon$ parameter can be used to control the second bias and obtain distributions with specific values of $\mathbb{E}\left[\Delta H_{1,2}\right]$. Next, we examine how Bias 1 influences the convergence of the MM and the CM to either of the possible causal factorizations.

**Observational case** Figure 4 shows the convergence behavior of the MM and CM model for different values of $\varepsilon$ in the observational case. Both the MM and the CM show a characteristic change in convergence. For $\varepsilon < 1$, the models converge to $c_1 = 0$, favouring to predict $X_1$. For $\varepsilon = 1$, the models do not converge to any direction, and for $\varepsilon > 1$, they favor predicting $X_2$. In summary, when $\mathbb{E}\left[\Delta H_{1,2}\right] = 0$ ($\varepsilon = 1$), the models do not display a convergence behavior, and if $\mathbb{E}\left[\Delta H_{1,2}\right] \neq 0$, both models favor predicting the variable with the lower entropy. These results are closely related to what Reisach et al. Reisach et al. (2021) report for continuous distributions and show that gradient-based structure learning algorithms can be susceptible to differences in the entropy of observed distributions in the observational case. Since many structure learning algorithms only consider observational data, it is important to understand whether their convergence behavior is due to learning conditionals or exploiting asymmetries in distributions.

---

[2]https://github.com/ec6dde01667145e58de60f864e05a4/CausalOptimizationAnon
[3]https://github.com/phlippe/ENCO
[4]Anonymized link: https://anonymous.4open.science/r/bias-based-causal-discovery-DC7A/

**Interventional case**   Bias 1 also exerts an effect on the convergence of both models in the interventional case. Specifically, for certain intervention-rate scenarios (here, $\lambda = 0.2$), Bias 1 is capable of reversing the direction toward which a gradient-based causal discovery model converges. As illustrated in Figure 5, the proportion of independent runs converging to $c_1 = 1$ increases with higher values of $\varepsilon$, mirroring the trend observed in the observational case. This highlights that, depending on the magnitude of $\Delta S_{1,2}$ induced by interventions, Bias 1 can still reverse the causal direction learned by gradient-based models.

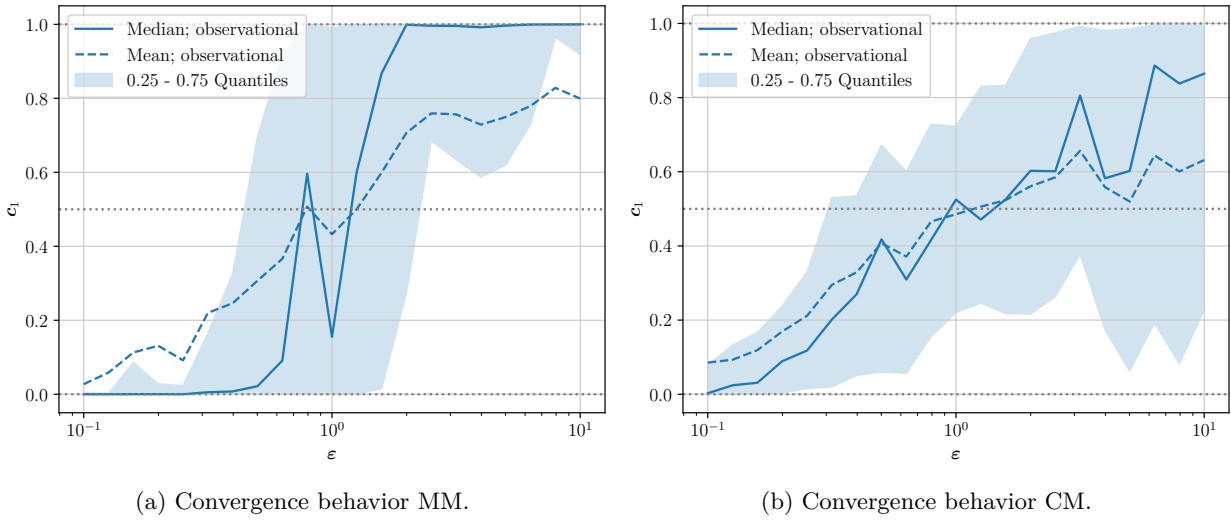

(a) Convergence behavior MM.

(b) Convergence behavior CM.

Figure 4: Bias 1 in the observational case: Value of $c_1$ after 300 epochs, plotted over $\varepsilon$ (log scale) for conditional model and marginal model. The statistics are based on 100 runs per value of $\varepsilon$.

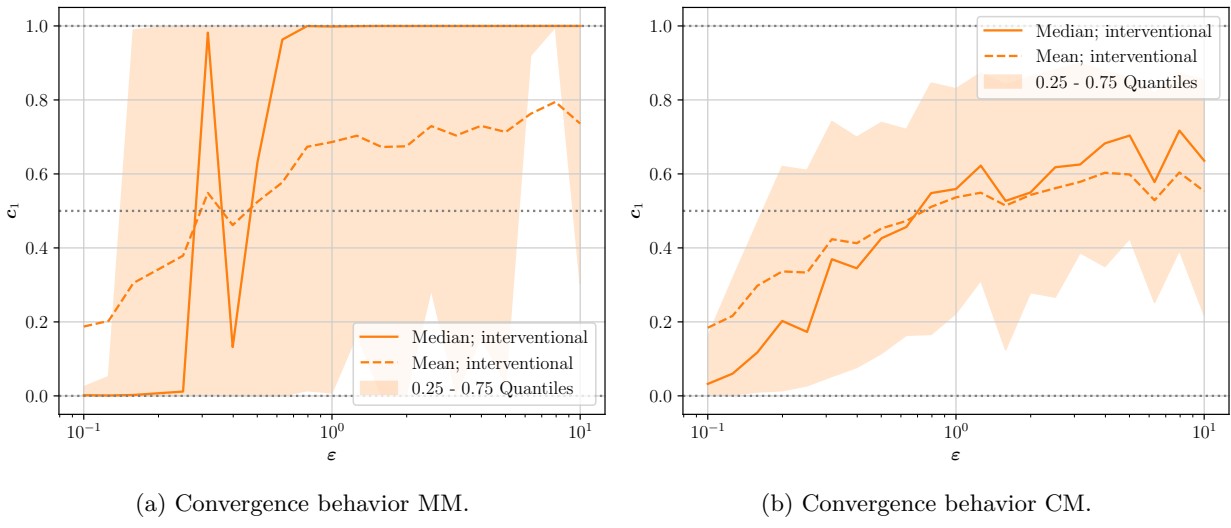

(a) Convergence behavior MM.

(b) Convergence behavior CM.

Figure 5: Bias 1 in the interventional case: Value of $c_1$ after 300 epochs, plotted over $\varepsilon$ (log scale) for conditional model and marginal model with $\lambda = 0.2$. The statistics are based on 100 runs per value of $\varepsilon$.

## 5.2   Bias 2: Marginal Distribution Shift Asymmetry

**Influence of $\lambda$ on distribution shift $\Delta S$**   The second bias exclusively impacts the interventional setting. To isolate its effect, we eliminate Bias 1 by setting $\varepsilon = 1$ throughout this section. Figure 3b presents empirical measurements of $\Delta S_{1,2}$ across varying values of $\lambda$. We observe that as $\lambda$ increases, $\Delta S_{1,2}$ strictly monotonically decreases. This monotonic behavior aligns with expectations from Section 3.2: with larger

$\lambda$, the frequency of second and third intervention types increases, gradually causing larger distributional shifts in $X_2$ on average. Additionally, achieving parity in distribution shifts ($\Delta S_{1,2} = 0$) requires a greater proportion of interventions on $X_1$ compared to $X_2$. Further analysis of how different intervention scenarios influence $\Delta S_{1,2}$ is available in Appendix Section A.4. Additionally, Appendix Section A.3, specifically Figure 11, demonstrates that employing cross entropy instead of Kullback-Leibler divergence yields a similar trend for $\Delta S_{1,2}^{CE}$.

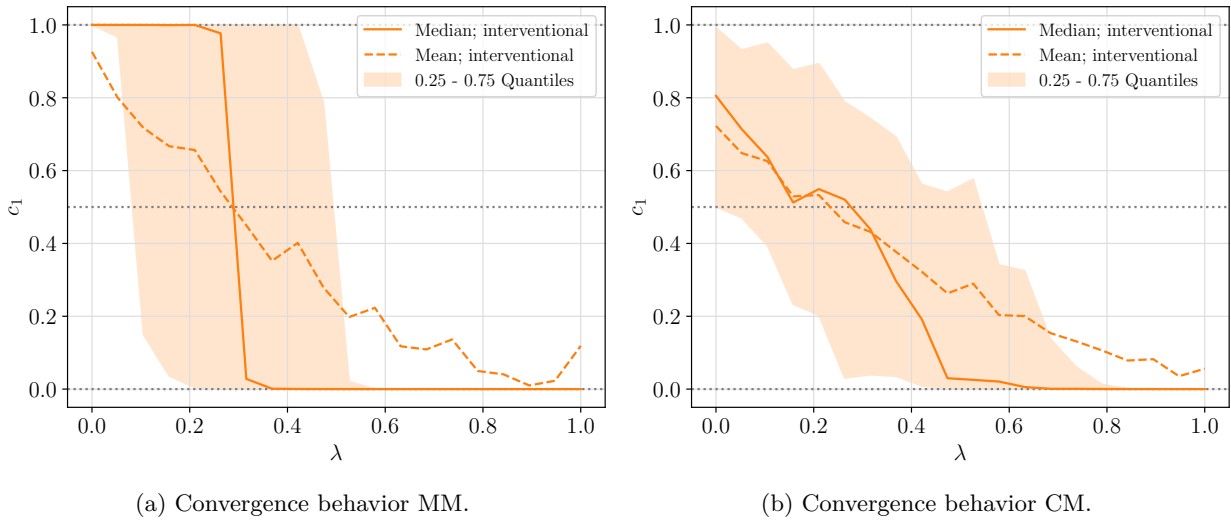

(a) Convergence behavior MM.                    (b) Convergence behavior CM.

Figure 6: Bias 2: Value of $c_1$ after 300 epochs, plotted over $\lambda$ for conditional model and marginal model. The statistics are based on 100 runs per value of $\lambda$.

**Gradual change of model convergence with $\lambda$**   Figure 6 shows the convergence of the MM and CM models under Bias 2 across different values of $\lambda$. Both models perceive the variable with the faster-changing marginal distribution (indicated by $\Delta S_{i,j} > 0$) as the independent variable. For low values of $\lambda$ (predominantly interventions on $X_1$), both models consistently converge to the factorization $X_1 \rightarrow X_2$ (i.e., $c_1 \approx 1$). Conversely, for high values of $\lambda$, the models gradually shift toward the opposite factorization, $X_2 \rightarrow X_1$ ($c_1 \approx 0$). This gradual transition in model convergence closely follows the pattern of changes observed in $\Delta S_{1,2}$, suggesting that the magnitude and direction of the distribution shift ($\Delta S$) directly steer the convergence behavior of the models. Importantly, the observed changes in $\Delta S_{1,2}$ alone are sufficient to fully reverse the direction toward which a gradient-based causal discovery model (whether learning conditional distributions or marginals only) converges, even in the absence of differences in entropy (Bias 1).

Lastly, consistent with observations from Bias 1, the MM again more effectively exploits the asymmetry, with its mean $c_1$ value closely tracking the changes in $\Delta S_{1,2}$ shown in Figures 3b and 11. Although the CM follows a similar general trend, it exhibits a flatter convergence curve. We further observe that the speed and stability of model convergence depend significantly on each model's ability to capture the underlying distribution before interventions occur. Figure 15 in Appendix Section A.6 illustrates this dependency.

### 5.3   Effects of Distributional Biases in Previous Work

In this section, we explore how the two biases influence other gradient-based causal discovery methods that consider the bivariate categorical setting.

**Meta-transfer objective**   We first investigate how the meta-transfer objective model proposed by Bengio et al. (2020) behaves with regard to Bias 1 and 2. As opposed to our illustrator models, this model first learns the probabilities of both possible causal factorizations $X_1 \rightarrow X_2$ and $X_2 \rightarrow X_1$ independently on observational data. Afterwards, both models achieve the same likelihood $\mathcal{L}$ on data sampled from the observational distribution, i.e. $\mathcal{L}_{X_1 \rightarrow X_2} = \mathcal{L}_{X_2 \rightarrow X_1}$, because both have learned the correct joined distribution.

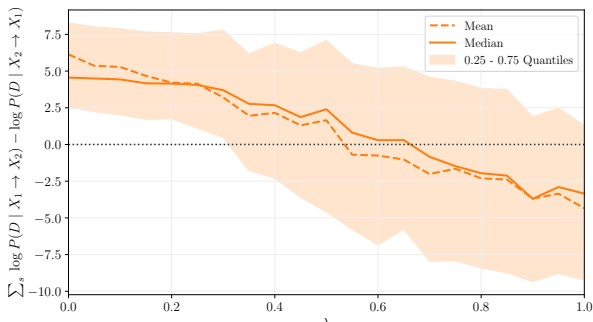 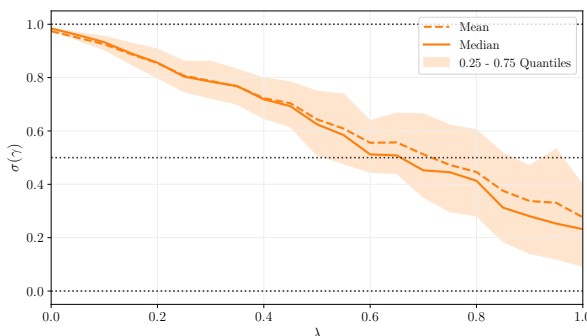

(a) The differences in log-likelihood between model $X_1 \to X_2$ and model $X_2 \to X_1$, summed over adaptation steps $s$. A value other than 0 means one model adapts faster than the other.

(b) The causal parameter $\sigma(\gamma)$ indicates the belief of the model that $X_1 \to X_2$ is the correct factorization (if $\sigma(\gamma) > 0.5$), or $X_2 \to X_1$ (if $\sigma(\gamma) < 0.5$)

Figure 7: Log-likelihoods (left) and derived causal factorizations (right) of the approach presented by Bengio et al. (2020), for different $\lambda$ values. For all runs, we set $\varepsilon = 1$.

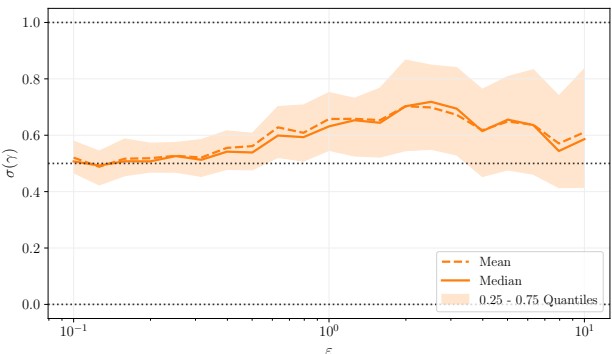

Figure 8: Causal factorization of the approach presented by Bengio et al. (2020), for different $\varepsilon$ values. The intervention rate was kept at $\lambda = 0.5$ for all $\varepsilon$ values.

In a second step, both pre-trained models are trained on interventional data. Their likelihoods now start to differ, until they have converged to the new joint distribution, because one factorization can adapt faster to the new interventional distributions than the other. The speed of adaptation depends on the distribution shifts described in Bias 2. Note that $\mathcal{L}_{X_1 \to X_2}$ and $\mathcal{L}_{X_2 \to X_1}$ are the accumulated marginal and conditional likelihoods.

While updating the models on interventional data, a scalar parameter $\gamma$, which encodes belief about the causal direction, is simultaneously optimized to minimize the following meta-objective, the *regret*:

$$\mathcal{R} = -\log\left[\sigma(\gamma)\mathcal{L}_{X_1 \to X_2} + (1 - \sigma(\gamma))\,\mathcal{L}_{X_2 \to X_1}\right]$$

Its gradient is $\frac{\partial \mathcal{R}}{\partial \gamma} = \sigma(\gamma) - \sigma(\gamma + \log \mathcal{L}_{X_1 \to X_2} - \log \mathcal{L}_{X_2 \to X_1})$, so the difference in likelihoods during model adaptation directly determines whether $\sigma(\gamma)$ converges to 0 or 1. In this paper, $\sigma(\cdot)$ is the sigmoid function.

Figure 7 shows the behavior of this approach for varying $\lambda$. Figure 7a shows the raw difference of the log-likelihoods of both models, summed over the first $s$ adaptation steps. It becomes apparent that the adaptation speed of each factorization is correlated with the distribution shift asymmetry presented in Bias 2. The convergence of $\sigma(\gamma)$ in Figure 7b behaves accordingly, and the predicted causal direction changes with increasing $\lambda$. This is in line with the results for our models presented in Figure 6 and shows the susceptibility of the approach by Bengio et al. (2020) to Bias 2.

We have further analyzed the susceptibility of the model proposed by Bengio et al. (2020) to Bias 1, which is shown in Figure 8. For this analysis, we set $\lambda = 0.5$ to obtain unbiased interventions. This means that $\sigma(\gamma)$ never reaches values near 1, as it does in the original paper where $\lambda$ is always 0. Although not as pronounced as for our models above, we observe a similar trend: the convergence to the causal hypothesis $X_1 \to X_2$ ($\sigma(\gamma) = 1$) generally increases for larger $\varepsilon$ values.

**ENCO**  We also investigate the susceptibility of the ENCO model by Lippe et al. (2022) to our proposed biases. As explained above, this approach splits the belief about a causal relation between two variables into a parameter $\sigma(\gamma_{ij})$ that encodes the belief that an edge between $X_i$ and $X_j$ exists in the SCM, and a parameter $\sigma(\theta_{ij})$ that encodes the direction of this edge. Figure 9 shows the behavior of those models for changing values of $\lambda$ and the edge $X_1 \to X_2$. We find that the parameter coding for edge existence, $\sigma(\gamma_{12})$, slowly declines from 1 to 0.5 but always stays above 0.5, while the parameter encoding edge direction, $\sigma(\theta_{12})$, stays near 1 for all values of $\lambda$. Together, this shows that the model is robust to the influences of Bias 2, as for all values of $\lambda < 1$, the model correctly predicts with a confidence greater than 0.5 that the edge $X_1 \to X_2$ exists. This robustness can be explained by looking at the gradients for both parameters. The gradient of the overall loss $\tilde{\mathcal{L}}$ for $\gamma_{12}$ is per Lippe et al. (2022) given as:

$$\frac{\partial \tilde{\mathcal{L}}}{\partial \gamma_{12}} = \sigma'(\gamma_{12}) \cdot \sigma(\theta_{12}) \cdot \mathbb{E}\left[\mathcal{L}_{X_1 \to X_2}(X_2) - \mathcal{L}_{X_1 \nrightarrow X_2}(X_2) + \lambda_{\text{sparse}}\right] \tag{12}$$

In the notation adopted by Lippe et al. (2022), $\mathcal{L}_{X_i \to X_j}(X_j)$ and $\mathcal{L}_{X_i \nrightarrow X_j}(X_j)$ denote the negative log-likelihoods (a different notation than for the meta-transfer objective above!) for variable $X_j$, if the edge $X_i \to X_j$ is included into the model ($X_j$ is conditioned on $X_i$) or not ($X_j$ is not conditioned on $X_i$), respectively. $\lambda_{sparse}$ is a regularization term to favor sparser causal graphs. Importantly, the gradient is artificially suppressed for the intervened-upon variable, since it is by definition independent. The formula explains why the model is robust to increasing values of $\lambda$: for $\lambda = 0$, in all samples the true causal direction $X_1 \to X_2$ holds, so including this edge should always yield a lower loss opposed to excluding it. As $\lambda$ grows, the causal relationship is present in fewer samples and the gradient is suppressed in more samples, so the signal to push $\gamma_{12}$ upwards (through gradient descent) becomes weaker, eventually moving to total uncertainty, $\sigma(\gamma_{12}) = 0.5$, at $\lambda = 1$.

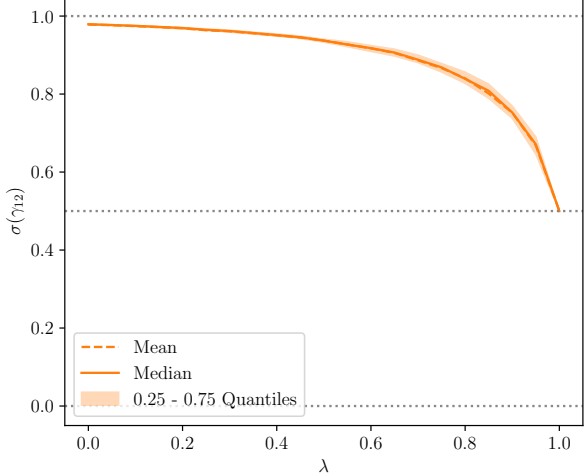
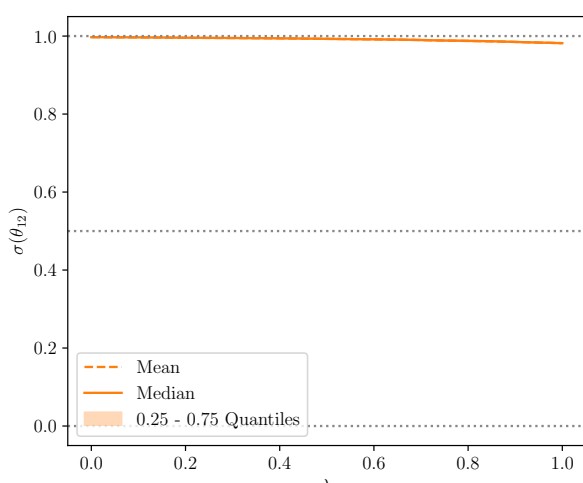

(a) The parameter $\sigma(\gamma_{12})$ encodes the belief of the model that an edge in the learned SCM, i.e. a causal relationship, exists between $X_1$ and $X_2$.

(b) The parameter $\sigma(\theta_{12})$ indicates the belief of the model that $X_1 \to X_2$ is the correct direction for the edge.

Figure 9: Edge existence belief (left) and edge direction belief (right) of the approach presented by Lippe et al. (2022), for different $\lambda$ values. For all runs, we set $\varepsilon = 1$.

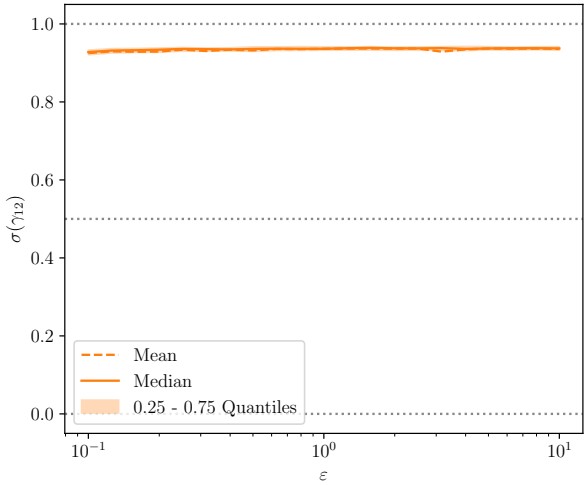 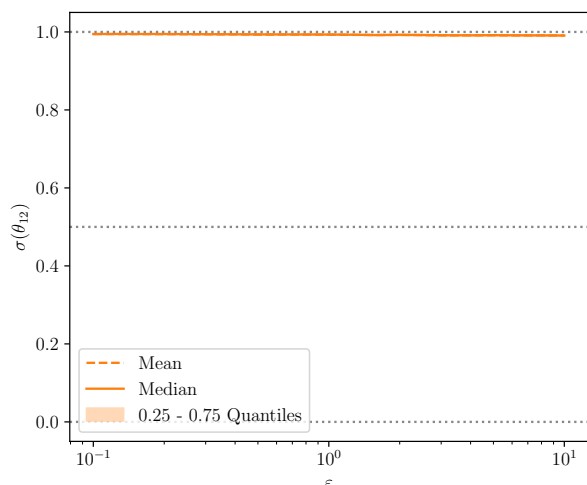

(a) The parameter $\sigma(\gamma_{12})$ encodes the belief of the model that an edge in the learned SCM, i.e. a causal relationship, exists between $X_1$ and $X_2$.

(b) The parameter $\sigma(\theta_{12})$ indicates the belief of the model that $X_1 \to X_2$ is the correct direction for the edge.

Figure 10: Edge existence belief (left) and edge direction belief (right) of the approach presented by Lippe et al. (2022), for different $\varepsilon$ values. For all runs, we use $\lambda = 0.5$.

The robustness of $\theta_{12}$ can similarly be explained by looking at the corresponding gradient:

$$\frac{\partial \tilde{\mathcal{L}}}{\partial \theta_{12}} = \sigma'(\theta_{12}) \Bigg[ p(I_{X_1}) \cdot \sigma(\gamma_{12}) \cdot \mathbb{E}\left( \mathcal{L}_{X_1 \to X_2}(X_2) - \mathcal{L}_{X_1 \not\to X_2}(X_2) \right)$$

$$- p(I_{X_2}) \cdot \sigma(\gamma_{21}) \cdot \mathbb{E}\left( \mathcal{L}_{X_2 \to X_1}(X_1) - \mathcal{L}_{X_2 \not\to X_1}(X_1) \right) \Bigg] \tag{13}$$

This gradient comprises two main components, each driven by interventions on one of the two variables. The relative influence of these components on the expected gradient is modulated by $\lambda$, i. e. the proportion of interventions. In the first term, $p(I_{X_1})$ is the probability of intervening on $X_1$, so has a high value when $\lambda$ is small and $X_1$ is intervened upon often. If $X_1 \to X_2$ is the true causal direction, modeling $X_2$ as dependent on $X_1$ leads to a better (smaller) negative log-likelihood than assuming independence, so $\mathcal{L}_{X_1 \to X_2}(X_2) - \mathcal{L}_{X_1 \not\to X_2}(X_2)$ is negative. Gradient descent then increases $\theta_{12}$. This gradient provides a direct signal for the correct causal direction.

The second term is weighted by $p(I_{X_2})$, which has a high value when $\lambda$ is large. If $X_1 \to X_2$ is true, then $X_1$ is invariant to interventions on $X_2$. Consequently, attempting to model $X_1$ conditioned on $X_2$ should offer no predictive improvement for $X_1$. Thus, the difference $\mathcal{L}_{X_2 \to X_1}(X_1) - \mathcal{L}_{X_2 \not\to X_1}(X_1)$ is expected to be zero or even positive, because the conditioning introduces noise. Since this entire term is subtracted in the gradient for $\theta_{12}$, it also contributes to pushing $\theta_{12}$ upwards.

Importantly, regardless of the value of $\lambda$, at least one of these mechanisms provides a signal (or at worst no signal, but never a contradicting one) for the correct orientation. When $\lambda$ is low, the direct causal signal from the first term dominates. When $\lambda$ is high, the invariance-based signal from the second term dominates. For intermediate values of $\lambda$, both types of interventions contribute.

Further, Figure 10 shows that both edge parameters are robust to Bias 1. This can also be read directly from the gradients above: the formulas do not compare likelihoods of variables themselves (which would be susceptible to Bias 1) but compare likelihoods of the same variable with or without edges added. This removes any influence of Bias 1.

Hence, the ENCO model is robust to both biases. The reason for this is that, as explained, ENCO does not find the correct causal graph by exploiting differences in distributions before and after interventions, but rather by probing which additions to the causal structure yield improvements to the likelihood of the data across interventions.

## 6 Conclusion

In this work, we inspect two different biases on the common choice of categorical distributions generated using a Dirichlet prior, and their influence on the behavior of gradient-based causal discovery methods in the bivariate case.

Bias 1 quantifies marginal distribution asymmetry and is measured through the difference in entropy between distributions. We show how it can be controlled, in the bivariate categorical setup, by a parameter $\varepsilon$. Bias 2 quantifies the distribution shift asymmetry of causally related variables under interventions. We analyze these changes and show how the intervention rate $\lambda$ can affect Bias 2, depending on the rate of interventions on each variable. This understanding of parameters $\varepsilon$ and $\lambda$ and their effect on the biases allows more rigorous testing of the robustness of causal discovery approaches.

We find that gradient-based methods can be susceptible to both biases – in particular if their learning involves a direct competition between different causal factorizations.

To illustrate this, we present two simple gradient-based models for causal discovery that transparently choose between conflicting factorizations. We find that neither these illustrator models, nor the meta-transfer objective approach by Bengio et al. (2020) are robust to the biases. Their predicted causal factorization changes when the biases are removed or reversed, even if the underlying causal structure that generates the data was unchanged. The approach by Lippe et al. (2022), on the other hand, is robust to both biases because it analyzes causal dependencies separately for each variable.

We believe that the presented biases and methods to control them will help other researchers to better analyze the robustness of their methods. We suggest to test methods on data that was generated for different values of parameters $\varepsilon$ and $\lambda$, and to consider their effect already in the design of new approaches.

## Ethics Statement

All experiments in this paper are conducted on synthetic data; thus, no issues regarding privacy or human subjects arise. Our work does not inherently facilitate harmful applications or negative societal impacts. The code is publicly available to enable transparency and reproducibility. The authors declare no competing interests.

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

# A    Appendix

## A.1    Definitions

To avoid ambiguity, we provide the common definitions of entropy and Kullback-Leibler divergence for discrete variables that we use throughout this paper, in particular also those for conditional distributions.

The entropy of a variable $X_i$ is defined as

$$H(X_i) = -\sum_{x_i \in X_i} P(x_i) \log P(x_i).$$

Conditional entropy is then defined as (Bishop & Bishop, 2024, Sec. 2.5.6)

$$H(X_i|X_j) = -\sum_{x_i \in X_i} \sum_{x_j \in X_j} P(x_i, x_j) \log P(x_i|x_j).$$

The Kullback-Leibler divergence of two distributions $P'(X_i)$ and $P(X_i)$ is defined as

$$D_{KL}\Big(P'(X_i) \,\big\|\, P(X_i)\Big) = \sum_{x_i \in X_i} P'(x_i) \log \frac{P'(x_i)}{P(x_i)}.$$

The conditional Kullback-Leibler divergence, consequently, is defined as (Cover, 1999, Eq. 2.65)

$$D_{KL}\Big(P'(X_i|X_j) \,\big\|\, P(X_i|X_j)\Big) = \sum_{x_i \in X_i} \sum_{x_j \in X_j} P'(x_i, x_j) \log \frac{P'(x_i|x_j)}{P(x_i|x_j)}.$$

## A.2    Relative Distribution Shift of Dependent Variable

**Theorem 1** (Distribution shift asymmetry under intervention on causal variable)**.** *Consider a bivariate causal model $X_1 \to X_2$ with distributions*

$$X_1 \sim P(X_1), X_2 \sim P(X_2 \mid X_1).$$

*An intervention on the cause variable $X_1$ changes its distribution from $P(X_1)$ to $P'(X_1)$ while leaving the conditional mechanism $P(X_2 \mid X_1)$ unchanged. Then, the induced marginal distribution $P(X_2)$ also changes to $P'(X_2)$. It holds that*

$$D_{KL}\Big(P'(X_2) \,\big\|\, P(X_2)\Big) \le D_{KL}\Big(P'(X_1) \,\big\|\, P(X_1)\Big). \tag{14}$$

*Proof.* Let $p$ and $q$ be any two probability distributions on a space $\mathcal{X}$, and let $K$ be any Markov kernel (i.e., a stochastic mapping) from $\mathcal{X}$ to another space $\mathcal{Y}$. Then the data-processing inequality for KL divergence (Cover, 1999, Theorem 2.8.1 and Eq. 11.145) states that

$$D_{KL}\big(pK \,\big\|\, qK\big) \le D_{KL}(p \,\|\, q)$$

Setting $p = P'(X_1), q = P(X_1)$, $K = P(X_2|X_1)$, marginalization by applying $K$ to $P'(X_1)$ and $P(X_1)$ (van Erven & Harremos, 2014) yields

$$D_{KL}\Big(P'(X_1)K \,\big\|\, P(X_1)K\Big) \le D_{KL}\Big(P'(X_1) \,\big\|\, P(X_1)\Big), \tag{15}$$

$$D_{KL}\Big(P'(X_2) \,\big\|\, P(X_2)\Big) \le D_{KL}\Big(P'(X_1) \,\big\|\, P(X_1)\Big). \tag{16}$$

$\square$

### A.3 Distribution Shift or Loss Shift

The following is based on the assumption that the model has already converged to a distribution before the intervention happens and changes that distribution.

For a cross entropy (CE) loss, the difference in loss shift becomes

$$\Delta S_{i,j}^{CE} \;=\; H(X_i') + D_{KL}(P_i'||P_i) \;-\; H(X_j') - D_{KL}(P_j'||P_j)\,.$$

The empirical Bias 2 based on CE loss is depicted in Figure 11. Its curve is very similar, but slightly flatter than the one based on Kullback-Leibler divergence (Figure 3b). Theorem 1 still applies if *Bias 1: Marginal Distribution Asymmetry* is not present, because then $H(X_i') = H(X_j')$.

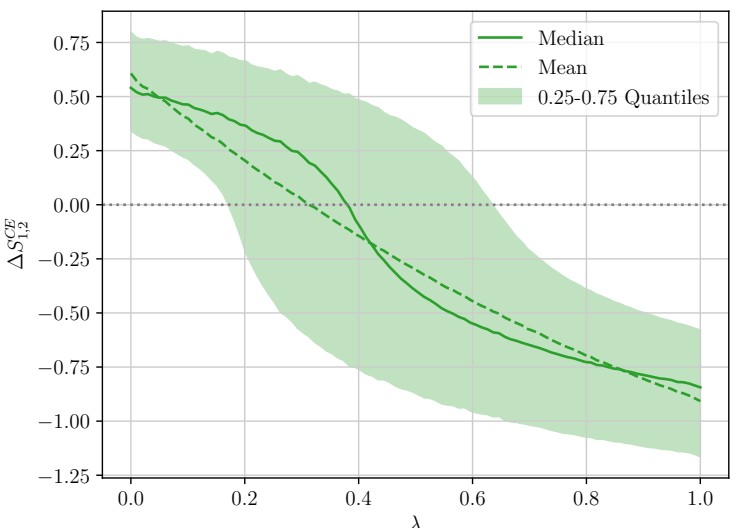

Figure 11: Plot of the empirical difference in cross entropy shift, as opposed to Kullback-Leibler divergence shift presented in Figure 3b. Interventions happened in random order.

### A.4 Intervention Cases in Practice

Figure 12 shows the ratios of intervention cases, as explained in Table 1, with increasing $\lambda$. Cases 1 and 2 result in causally related variables (see Figure 1), while cases 3 and 4 do not. Consequently, the proportion of causal information in the data sinks with increasing $\lambda$ – in fact linearly, because the sum of case 1 and case 2 ratios, and likewise case 3 and case 4 ratios, is linear. This is consistent with the convergence findings of the marginal and conditional model as well as the related work in Section 5.3. Figure 13 and 14 provide an empirical scatter plot of distribution shifts for different intervention cases in the bivariate categorical setup described in Section 4. Figure 13 shows the marginal distirbution shifts. In case 1, $D_{KL}(P_1'||P_1)$ is always smaller than $D_{KL}(P_2'||P_2)$, consistent with Theorem 1. Overall, the observations for both marginal and conditional distribution shift empirically verify the descriptions in Table 1.

### A.5 Model Details

This section details the hyperparameters and initializations for the MM and CM models. In both models, the parameters of $\mathbf{W}$ are initialized using a Kaiming Uniform distribution, and the parameters $z_1$ and $z_2$ were initialized to 0.5 each. The parameter $\mathbf{i}$ of the MM was initialized to $\mathbf{1}_K/K$. This ensures the value is normalized to 1, as the input in the case of the CM, and additionally represents maximal uncertainty about the data distribution. Both models are trained for 300 epochs with 32 batches of size 128 in each epoch, using the Adam optimizer (Kingma & Ba, 2015) with a learning rate of 0.1. The temperature parameter of the Gumbel-Softmax was fixed to $\tau = 2$ throughout all experiments.

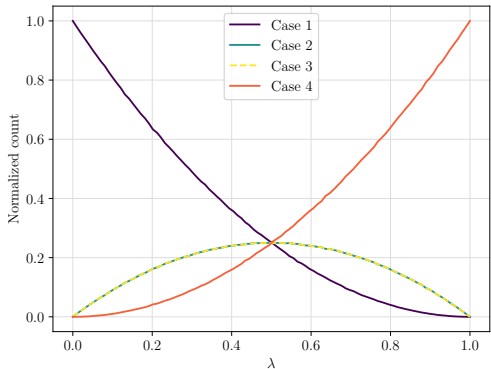

Figure 12: Plot of the ratio in which intervention cases 1, 2, 3 and 4 from Section 3.2 occur depending on $\lambda$, when interventions happen in random order. The ratios sum to 1 for each $\lambda$.

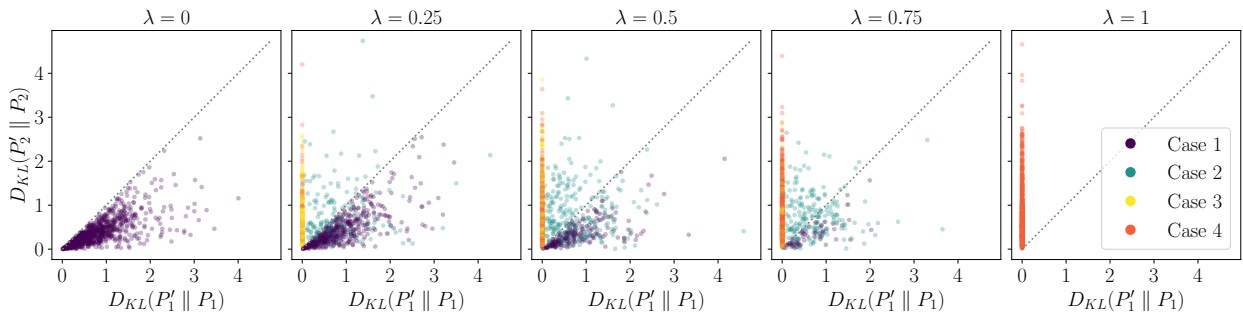

Figure 13: Scatter plot of distribution shifts of $P(X_1)$ and $P(X_2)$ for different $\lambda$ values. Cases 1, 2, 3, 4 of interventions are described in Section 3.2. Interventions are performed in random order. The diagonal represents the case $\Delta S_{1,2} = 0$. The ratios of points per case in each plot can be inferred from Figure 12.

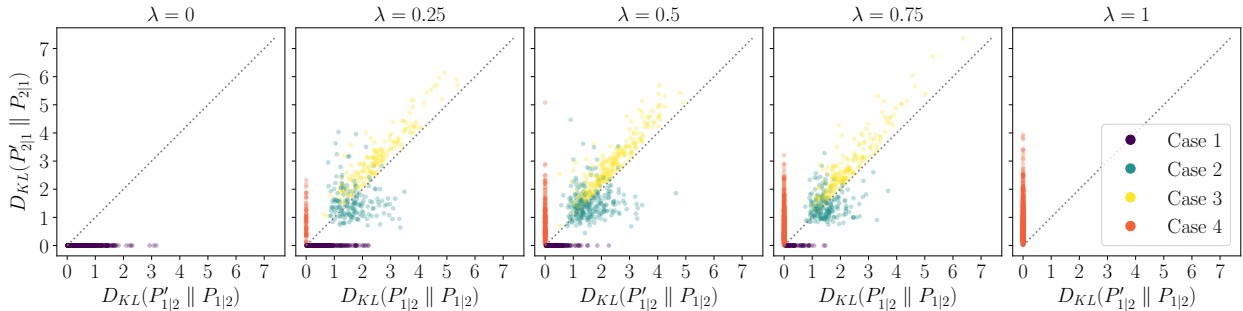

Figure 14: Scatter plot of distribution shifts of $P(X_1|X_2)$ and $P(X_2|X_1)$ for different $\lambda$ values. Cases 1, 2, 3, 4 of interventions are described in Section 3.2. Interventions are performed in random order. The diagonal represents the case $\Delta S_{1|2,2|1} = 0$. The ratios of points per case in each plot can be inferred from Figure 12.

## A.6 Fast Versus Slow Interventions

Figure 15 shows how the influence of Bias 2 differs depending on how frequently an intervention is performed. The more batches the model is trained on before another intervention happens, the earlier the model switches from $c_1 > 0.5$ to $c_1 < 0.5$ with respect to $\lambda$.

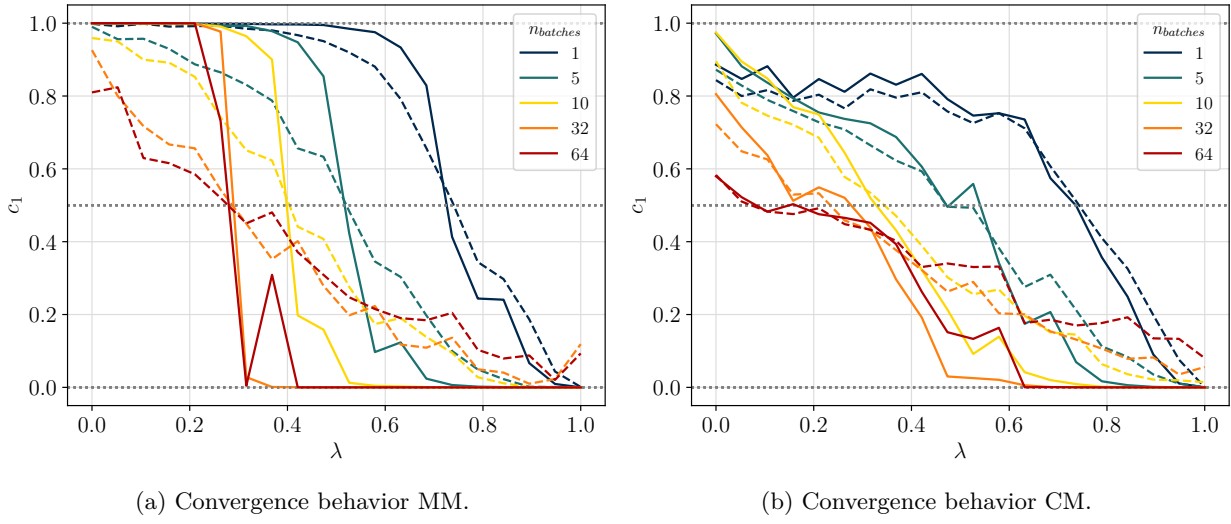

(a) Convergence behavior MM.

(b) Convergence behavior CM.

Figure 15: Value of $c_1$ after 300 epochs, plotted over $\lambda$ for conditional model and marginal model. The statistics are based on 100 runs per value of $\lambda$.

