# OpenReview forum: "Effects of Distributional Biases on Gradient-Based Causal Discovery in the Bivariate Categorical Case"
_TMLR — Withdrawn by Authors_

### Review · Reviewer_bCxE · 2025-07-31

**Summary Of Contributions:**

This manuscript examines the problem of identifying causal relationships between two categorical variables with interventions.  A primary focus is on the impact of the properties of the two variables, namely when one has a greater marginal entropy than the other and the impact on their distributional shifts are variable.  This is analyzed by an empirical model, along with evaluating how existing approaches handle these different data challenges.

**Audience:**

Yes

**Claims And Evidence:**

No

**Requested Changes:**

Overall, the manuscript should be rewritten for clarity, and to better explain the motivation for the new modeling approaches, including details on why this approach is chosen compared to standard practice.

The insights need to be clarified, and it needs to be more clearly distilled how practitioners and methodologists can translate these approaches to clearly understand robustness in novel applications.

There needs to be at a minimum a healthy discussion of the impact on larger systems, and how these issues might propagate.

**Strengths And Weaknesses:**

### Strengths:

There is a deep consideration of the particular data properties and the way they interact with causal discovery algorithms.

### Weaknesses:

Overall, I think that this manuscript has some issues with clarity and precision of the developments, along with elucidating a clear motivation.  In particular, a new framework based on the BDe is first introduced, which seems to struggle with some of issues here, although they can be mitigated, before evaluating existing approaches.  The authors then note that ENCO seems to handle this data properties pretty well, making it unclear why there was an initial introduction of a different model.  I cannot fully discern what the gap was that the new model was supposed to be solving.  I think that a careful recrafting of the manuscript to make it clearer that this is primarily an evaluation how these challenges impact existing approaches and a new approach would enhance the clarity of the manuscript.

There are myriad issues with the notation and claims, which while all minor, add up to it being challenging to fully capture what the authors are intending.  Some particular issues:
* Definition 2: $S_i$ is not "distribution shift" itself as KL is a measure of difference between two probability distributions, so this should be instead described as a measure of distribution shift.  There are many different distribution shifts that could be mapped to the same $S_i$.  Then $\Delta S_{i,j}$ would be more accurately described the the difference between the measures of distribution shift in the two distributions.
*"We assume continuous interventions, where a new intervention is applied immediately upon
undoing a previous intervention." -> this is not describe or explained what is meant by this, which needs to be much more formally defined.
*Figure 1: "In case 2, the underlying conditional P(X2|X1) is restored." This is dependent on the way this is done, etc., and should not be stated so casually. If the scenario on the right is correct where there is independence between X1 and X2, then the coupling would not be established by intervening on X1.  This is the same in Table 1, where case 2 states that "P2 also changes arbitrarily as it was arbitrary before intervention" But it wouldn't change arbitrarily, as is it would be depending on the coupling P(X2|X1) if indeed that is restored.  It needs to be clarified what is meant by decoupling, and what the authors mean by "arbitrarily."
* Table 1, Case 4.  The authors state that "P2|1 changes arbitrarily."  However, it would simply just change to the marginal, all in the same way regardless of X1, which does not feel appropriately captured by "arbitrarily."  Implying the marginal changes implies a causal relationship between them.
* The motivation for coupling W in (8) and (9) is unclear, along with using this form for the gumbel-softmax.
* "From a causal discovery point of view, the structural parameters c1,c2 can be viewed as the causal factorization learned by each model: c1 >0.5 or c1 <0.5 denotes the case where X1 causes X2 or vice versa."  This is not clear that this is fully appropriate, as the the W is shared and it's not clear that the states of X have similar meanings.  Suppose c1 was at .1, but the W_1 (state 1 column) was very large but only used by X_1 and not by X_2.  Under this scenario, the value of c_1 could be fairly meaningless for determining the causal direction (which seems highlighted by the results below).
* The modeling formulation here is not completely clear, as it's not clear that it matches existing practice, nor is it clear to me how it was designed to capture the correct causal inference solution.
* There is no comparison to standard causal inference approaches, which compare the plausibility of two causal models.  This choice needs to be heavily discussed.
* Observational case: "Since many structure learning algorithms only consider observational data, it is important to understand whether their convergence behavior is due to learning conditionals or exploiting asymmetries in distributions."  It is not clear what insights are provided by this statement or method.  What are the authors trying to claim about observational cases from their results, especially given they are not doing an observational case?
* Overall, the conclusions from the experimental results are not clearly distilled enough.  Based on the results, it seems like the conclusion is to use ENCO, and to try to balance which variables receive interventions.
* In the conclusions, the authors state "We believe that the presented biases and methods to control them will help other researchers to better analyze the robustness of their methods." If that is the goal, the manuscript should provide a clear guide for how researchers should do this.

Overall, the analysis here is limited to the bivariate case.  As probabilistic models, which don't obviously suffer from the challenges of gradient-based causal discovery methods, are quite feasible here, the authors need to explain how this impacts larger scale systems and experiments, and how the insights are transferrable to those cases.

---

### Review · Reviewer_gu5h · 2025-08-08

**Summary Of Contributions:**

The authors present two biases that can impact differentiable causal discovery methods. They show empirically with two toy models that the biases can have a drastic effect. They also apply their analysis to two existing causal discovery methods.

**Audience:**

Yes

**Broader Impact Concerns:**

No concern

**Claims And Evidence:**

Yes

**Requested Changes:**

Overall, an interesting paper that is well-written and can have an impact on the community. However, I think that material could be added to make the case stronger and more general. The authors have identified two biases and shown empirically that it can affect two differentiable causal discovery methods. There are no theoretical results. It is thus not clear to me how the results would apply to different settings (e.g., for other differentiable causal discovery methods, for continuous data, etc). Furthermore, they mostly observe the effect on the Bengio et al., 2020 method, which departs from many classical causal discovery settings.

Overall, my requested changes are mostly to add explanations and some experiments.

- **Theoretical insights.** _[Critical: Add a discussion | Would strengthen the work: Add experiments with another differentiable causal discovery method.]_ Are there any theoretical insights that could be added or, at least, an intuition? I think this is important; otherwise, the evaluation should be more extensive. Right now, it is not clear if the results are generalizable to other settings. For example, would the bias be present for another common causal discovery method such as DCDI (Brouillard et al., 2020)? Also, does the choice of optimizer or the use continuous data have an influence on the bias? Can we be sure it would not affect score based causal discovery methods?

- **Adding a control to experiments.** _[Would strengthen the work]_ I believe that a “control” condition should be added to clearly show that the effect is specific to differentiable methods. Non-differentiable methods, such as IGSP or GIES, could be used. I am thinking more particularly about the observational case of bias 1, when using different values of $\epsilon$ in the generative model, it seems to me that it could also lead to a problem for other methods. High values of $\epsilon$ lead to almost deterministic distribution, and low values decrease the signal-to-noise ratio (all methods are affected by that, see Montagna et al., 2023).

- **How to solve this problem?** _[Critical: Add a discussion]_ Reading the paper, I wondered if there are easy ways to prevent the bias. What happens when data are normalized as it was suggested in Reisach et al., 2021? Could the authors show empirically how it affects their conclusion? For the second bias, could a resampling of the data alleviate the bias?

- **Linear Gaussian case.** _[Would strengthen the work]_ I was thinking of the linear Gaussian case since I am more familiar to it. Suppose that we have $X_1 \rightarrow X_2$ and that $X_1 \sim N(0, 1)$ and $X_2 = \alpha X_1 + \beta E$ where $E \sim N(0, 1)$. Then, ways to increase $\Delta H_{1, 2}$ is to decrease $\alpha$ and/or $\beta$ and vice-versa. If someone change only $\beta$, then it corresponds to changing the signal-to-noise ratio and as said previously, it affects all methods. On the other hand, if $\alpha$ is changed, a standardization should remove the bias. I would be interested to see if the authors thought more about the continuous setting.

- **Bias 1, Observational case.** _[Would strengthen the work]_ In the observational case, bivariate graphs are not identifiable… All causal discovery methods would fail in general to recover the right direction. Could the bias effect also be shown for simple graphs that are identifiable, such as a v-structure? The authors mention that for bias 1 in the observational case, the results are closely related to what Reisach et al., 2021 reported. Could the authors make a stronger link with Reisach et al., 2021? Can it be seen as a particular case of Marginal distribution asymmetry? For example, can it be shown that for a linear model, the varsortability is similarly explained by the asymmetry?

- **Bias 1, Interventional case.** _[Would strengthen the work]_ Out of curiosity, the reported results are when $\lambda = 0.2$. Was the bias effect not observable for other values of lambda, such as $\lambda = 0.5$? If that is the case, then does using a resampling procedure to equilibrate the number of interventions on each variable mitigate the bias effect?

- **Clarify the setting.** _[Critical]_ While the paper is overall clear, I am not totally sure I understand the way the data are generated with respect to interventions. The authors say that they assume continuous interventions — this is not a classical term. Why do the authors use the setting from Bengio et al.,2020, which is not common? Could the authors clarify the setting in the paper and explain their choice?

- **Adding Priol et al., 2021.** _[Critical]_ For the work Bengio et al., 2020, the study from Priol et al., 2021 should at least be cited (and optimally discussed): it also shows bias based on convergence rates from stochastic optimization.


- **Typos** _[Critical]_:
    - p.2, point 2: “case,and”, a space is missing.
    - p.3 and others: Usually, using expressions such as “e.g.” and “i.e.”, a comma is commonly used right after.
    - p.6 Eq.3, p.8 Eq.10 and others: The punctuation for several equations is not respected. Periods should be used when sentences end.
    - p.7, 2nd paragraph: “from Dirichlet prior Dirichlet”. Also, “as for P(X1) here.”
    - p.9 first paragraph: “Biases” should not be capitalized.
    - p.9, last paragraph: “Reisach et al. Reisach et al.”
    - p.14: “so has a high value” sounds abrupt.

**Strengths And Weaknesses:**

__Strengths:__
- Interesting and well-written paper that is clear.
- The impact on the community can be important since differentiable causal discovery has become common.

__Weaknesses:__
- The results are empirical, and there are no theoretical results. For example, it is not clear to what extent the bias would affect other methods.
- The experiments could be more extensive.

---

### Review · Reviewer_Fiz5 · 2025-08-13

**Summary Of Contributions:**

This paper provides some evidence that the approach to causal discovery with interventions adopted in Bengio's et al. (2020) is not suitable for causal discovery. For example, the Markov equivalence class of the graph $X_1 \to X_2$ contains two graph. Bengio et al. use interventions to distinguish between the two graphs; in this paper, the empirical results suggest that:

- if the "interventional" dataset contains more interventions over $X_1$ than over $X_2$, the model will tend to output the causal direction $X_1 \to X_2$,
- if the "interventional" dataset contains more interventions over $X_2$ than over $X_1$, the model will tend to output the causal direction $X_2 \to X_1$.

Given that the ratio of interventions of one variable over the other should not depend on the causal graph (which is unknown), the approach in Bengio et al. does not appear to be suitable for causal discovery. Empirical evidence in support of this hypothesis, although not overwhelming, is the main contribution of this work.

**Audience:**

No

**Broader Impact Concerns:**

I don't see any relevant concern in terms of broader impact.

**Claims And Evidence:**

No

**Requested Changes:**

1. As elaborated in the Weaknesses section before, my main concern is that the experimental set up is not suitable for demonstrating that Bias 1 and Bias 2 defined in the paper steer gradient-based causal discovery algorithms in undesirable ways. Unless the authors can provide strong evidence for this, I don't see a straightforward way to support their claims. Thus, if possible, I require the authors to provide more convincing discussion and experiments on why the learners of Section 5.1 and 5.2 are actually predicting the causal direction via the parameter $c_1$.
2. In Section 5.3, the authors claim that the empirical results of Fig. 7 are explained by Bias 2: this is not obvious from their discussion. The interpretation is credible only if the connection with Table 1 (as described in the "Strengths and Weaknesses" section above) is explicit, which is not the case in the manuscript. I ask the authors to make this connection explicitly. In order to strengthen these results, it would also be necessary to provide a theoretical proof of the claims of Table 1, which I believe is currently missing
3. As previously discussed, the empirical evidence does not convincingly support that Bias 1 and Bias 2 steers gradient-based causal discovery: in Section 5.3 this is particularly evident in reference to the experiments with the ENCO architecture. Can the authors show that ENCO-like gradient-based methods systematically revert their predictions due to the highlighted biases?

**Strengths And Weaknesses:**

# Weaknesses

__My main concern is that the interpretation of many empirical results of the authors feels arbitrary and not supported by the empirical evidence__. In particular, I believe that the *causal* interpretation of the empirical results on the MM (marginal model) and CM (conditional model) learners is not in scope, and does not suggest that gradient-based causal discovery is inherently biased by the *Marginal distribution* and *Marginal distribution shift* asymmetries.

Sketching the argument (elaborated later on): the interpretation of $c_1$ learnable parameter being the causal direction privileged by the learner feels arbitrary — and thus, the main conclusion that causal discovery with gradient-based methods is biased, is not well supported. The point is that $c_1$ is set by the learner such that the distribution of the *lowest entropy* random variable is learnt: this explanation does not require any *causal* interpretation of the data-generating process or the prediction. In all the experimental settings of sections 5.1 and 5.2 the model is given the choice to learn only one distribution among two. Generally speaking, a *high entropy* distribution is closer to a uniform distribution. The key point is that predicting the outcome of a uniform distribution is generally harder than predicting the outcome from a *low entropy* distribution. So, if the model can only learn one distribution, it will privilege the low-entropy one. Reading the empirical results through this lens provides an interpretation of the empirical results of sections 5.1 and 5.2 that is completely detached from the *causal* interpretation the authors give to the learnable parameter $c_1$. I’ll elaborate in detail below.

---

## Weaknesses in the analysis of Figure 4

About the results in Figure 4, a very high $\epsilon$ in the Dirichlet parameters implies a distribution of $X_2$ where most of the mass is concentrated on one category: e.g., for $X_2$ Bernoulli $p(X_2)$ concentrates all the mass on one value, say $0$ or $1$. On the other hand, consider the bivariate $X_1$ with $p(X_1=1)=0.5$. These categorical distributions could result from the experimental setting of the paper with $K=2$ and $\epsilon$ large enough, which, according to the authors (end of page 6), makes $p(X_1)$ *more uniform* (with higher entropy) than $p(X_2)$.

Say that, under these conditions, I can use only one parameter to learn the two bivariate distributions (which would require two parameters to be described): then, the best strategy to minimize the loss is to decide modelling the probability of $X_2$, i.e. set $c_1 = 1$. This is because the following two cases are possible:

- **Case 1**: if the learner decide to model the probability of $X_1$ with its single parameter this means it sets $c_1 = 0$ and hence, for $X_2$ the model assigns uniform probability to both outcomes. The accuracy of such a model in predicting $X_2$ outcomes is  $50$%. On the other hand, the model actively learns the distribution of $X_1$: being this a fair Bernoulli, correctly modelling the probability results in a prediction accuracy over $X_1$ outcomes of 50%.
- **Case 2**: If the learner decides to model the probability of $X_2$ with the single learnable parameter, instead, it sets $c_1=1$: this would result in a model that predicts $X_2$ with 100% accuracy given enough training data. Instead, with no learnable parameter left, the model assigns uniform probability over all possible outcomes of $X_1$, which will result in 50% prediction accuracy for $X_1$.

It is clear from the example that the logic to set optimal $c_1$ to minimize the prediction error is that the random variable with higher entropy (*more uniform*) can never be modeled as well as the random variable with lower entropy: then, to minimize the loss it’s better to use your parameters to model the distribution of the random variable with the lower entropy. This explains the results in Figure 4: for $\epsilon >> 1$, $X_1$ is the variable with higher entropy, and thus imply $c_1=1$, in agreement with Fig. 4. Viceversa, $\epsilon << 1$ imply that $X_2$ has higher entropy, and thus $c_1=0$, again in line with Fig. 4. __This interpretation is based on the principle that the model learns parameters that minimize the loss function, while it does not require that  $c_1$ encodes the causal direction, as claimed by the authors__: in fact the plots in figure 4 should result from any learning problem where you have a little number of learnable parameters, and two distributions to learn, one describing a high entropy random variable, the other a low entropy random variable. In this sense, __I think that the paper's discussion or experiments don't adequately support that $c_1$ has this causal meaning. Hence, it is not well supported that bias 1 steers the gradient-based causal discovery learner.__

---

## Weaknesses in the analysis of Figures 5, 6

Figures 5 and 6 can be interpreted with a logic that is similar to the above, which reinforces my belief that $c_1$ does not have the causal meaning assigned by the authors, and thus that the conclusion that biases steer gradient-based causal discovery is not well supported.

In Figure 6, we observe that the model tends to learn the distribution of the random variable that is less intervened on. Now, consider the case of interventions occurring mostly on $X_1$. According to Table 1, Case 1, such an intervention induces a change in $P_1$ that is greater than a change in $P_2$. Now, given that each batch in the dataset is sampled from a (potentially) different distribution, between each batch the distribution of $X_1$ will change more than that of $X_2$: then, the empirical distribution of $X_1$ in the interventional dataset is associated with higher entropy than the empirical distribution of $X_2$ in the dataset. We previously explained why, under these conditions ($X_1$  observations with higher entropy), the optimal choice of parameters for a learner is to set $c_1=1$: this is in line with what we observe in figure 6a (I believe a similar interpretation applies for Figure 6b). Note that this reasoning also explains the observations in Figure 5.

__For this reason, as for the discussion on the experiment analysed in Figure 4, I believe that the author's interpretation of $c_1$ as the learned causal direction is not supported by their discussion and their experiments__

---

# Strengths and further notes

I will further elaborate on the "summary of contributions" above. Compared to the analysis of Section 5.1 and 5.2, the explanation the authors give for the results observed in section 5.3 is more convincing: Bengio et al.’s architecture rely on how *fast* the learner can adapt to the distribution after intervention: if you pair this with how the conditional distribution changes under an intervention in cases 1 and 3 of Table 1, the empirical observation aligns with the theoretical intuition of Table 1. In this regard, however, I have some notes:

1. In Section 5.3, the authors claim that the empirical results of Fig. 7 are explained by Bias 2: this is not obvious from their discussion. The connection is clear only if the description of the distribution changes of Table 1 is explicit, which is not the case in the manuscript.
2. This contribution is interesting, but not itself sufficient for a publication. For instance, Lippe et al. and Ke et al. adopt a different learning strategy, i.e. that of learning the probability of a graph given the data. As the experimental results show (Fig. 9) that this approach is robust to Bias 2. Then, the contribution amounts to one experimental setting that suggests (I believe rightfully so) that Bengio et al.’s approach is not suitable, but the conclusion does not generalize to gradient-based causal discovery

---

### Note · Authors · 2025-08-25

**Comment:**

We thank the reviewers for their detailed and thorough feedback on our manuscript. We found it very helpful and agree that the paper would benefit from the following changes:
- Analyzing further related gradient-based causal discovery methods to to improve generalization of our claims.
- Experiments beyond the bivariate categorical case, e.g., with more variables and/or with continuous variables.
- Basing the analysis entirely on existing approaches instead of using our own models. Our models are a simplified version of the approach by Bengio et al. and are intended to show how _not_ to do causal discovery (precisely because $c_1$ fails to reliably encode causality) and why this is the case. However, we realize that they cause some confusion.

We feel that these changes constitute a major rework of the existing manuscript and will thus result in a different, new, paper. Hence, we respectfully withdraw the current submission. We are however grateful for the reviewer's work and suggestions as they will help us in improving the paper.

**Withdrawal Confirmation:**

I have read and agree with the venue's withdrawal policy on behalf of myself and my co-authors.